## Perspective

decoloniality; coloniality; migrants; refugees; communication

**Corresponding author:**
Gian-Louis Hernandez;
Email: g.hernandez@hva.nl

# Mental health care for migrants in the Netherlands: A decolonial perspective

Gian-Louis Hernandez[1,2] (iD), Melanie de Looper[1,3], Sabine Braun[4], Graham Hieke[4], Demi Krystallidou[4], Julia van Weert[1] and Barbara Schouten[1]

[1]Amsterdam School of International Business, University of Amsterdam, Amsterdam, The Netherlands; [2]Amsterdam School of Communication Research, Amsterdam University of Applied Sciences, Amsterdam, The Netherlands; [3]Tranzo, Scientific Center for Care and Wellbeing, Tilburg University, Tilburg, The Netherlands and [4]Centre of Translation Studies, University of Surrey, Surrey, UK

## Abstract

This study addresses the mental health needs of refugees and migrants in the Netherlands, highlighting the urgent public health challenges they face. Unique psychosocial hurdles, exacerbated by cultural dislocation, language barriers and systemic inequalities, hinder their access to quality mental healthcare. This study explores how coloniality intersects with mental healthcare access, using a decolonial framework to challenge stereotypes and assumptions that marginalize migrant voices. Through semi-structured interviews with migrants and language service providers, this research reveals the complexities of navigating the mental healthcare system. Findings reveal that temporality, professionalism and language barriers are key issues in migrants' mental healthcare journeys. We advocate for systemic changes that prioritize migrant perspectives. Ultimately, this study aims to inform policy and practice to enhance mental health services for migrant populations in the Netherlands and contribute to the broader dialogue on decolonization in mental health.

## Impact statement

The mental health needs of migrants in the Netherlands are often overlooked, highlighting a critical gap in both the understanding and provision of mental healthcare services. This article presents a decolonial perspective on the experiences of migrants navigating the mental healthcare system, emphasizing the importance of cultural competence and structural equity. By analyzing the barriers faced by migrants, such as low language proficiency and cultural misunderstandings, this study offers valuable insights into the systemic inequities that persist within mental health frameworks. The findings underscore the necessity for inclusive and empathetic mental health services that are tailored to the unique needs of migrant populations. Recommendations include enhancing training for healthcare providers in cultural humility, advocating for policy changes, prioritizing funding for migrant mental health initiatives and fostering community engagement to build trust and collaboration. The impact of this article extends beyond the Netherlands, as it provides a framework for understanding the intersection of coloniality and mental health in diverse global contexts. By sharing best practices and lessons learned, this study encourages mental health researchers and practitioners worldwide to adopt a decolonial lens, ultimately contributing to more equitable and effective mental health care for marginalized populations. The insights gained from this research can inform future studies and interventions, promoting a more inclusive approach to global mental health that recognizes and addresses the complexities of migration and mental health care.

## Social media summary

New research reveals critical gaps in mental health support for migrants in the Netherlands. A decolonial lens shows how language, culture and historical power dynamics create systemic barriers to care. Time to center migrant voices and build truly inclusive mental health services. #MentalHealth #Migrants #Decolonization.

## Findings

1. Temporality: Uncovers provider–patient power dynamics, revealing the contrast between expected punctuality and the experiences of migrants. This analysis challenges cultural stereotypes about time, introducing the concept of "Dutch time" from migrant perspectives.

Moreover, the study illuminates strategies employed by migrants to exert agency through temporal disruptions, such as emotional outbursts.

2. Professionalism: The analysis juxtaposes Western concepts of proficiency with migrant perspectives, which emphasize intimacy and kindness as essential to professionalism. This paradigm shift underscores the need to reevaluate institutional motivations and highlights migrants as experts in defining their own care preferences.

3. Language choice: The study sheds light on the colonial nature of English as a lingua franca and the need for a nuanced understanding of language dynamics. This revelation is critical for dismantling colonial structures in healthcare.

Practically this study suggests systemic changes in mental healthcare. Improved expectation management and a nuanced approach to language barriers can enhance the quality of care. In addition the study informs healthcare providers to better navigate language structures linked to coloniality. Theoretically this research pioneers a decolonial framework for mental health unveiling how colonial structures are entrenched within healthcare systems. It provides insights that can reshape practices to ensure equitable access to mental healthcare. In conclusion this article's decolonial perspective has the potential to spark transformative change in the mental healthcare landscape of the Netherlands and beyond.

## Introduction

Addressing the mental health needs of refugees and other migrants in the Netherlands is a pressing public health concern that requires urgent attention. Migrants often face unique psychosocial challenges, exacerbated by cultural dislocation, language barriers and systemic inequalities within the healthcare system (Schouten et al., 2022). These challenges can significantly hinder their access to and quality of mental healthcare, leading to poorer health outcomes and diminished quality of life. Research indicates that migrants are less likely to seek professional mental health support compared to nonmigrant populations despite experiencing higher rates of mental health issues (Lindert et al., 2008; Rousseau and Frounfelker, 2019). These inequalities can be traced back to, for example, historical and colonial relationships between nation-states that spur global economic inequality and larger migration flows, with a marked impact on access to mental healthcare (Mills, 2017).

By employing a decolonial framework – a framework that aims to dismantle the enduring effects of colonialism and elevate the voices of marginalized populations (Enck-Wanzer, 2012; Escobar, 2013; Bhatia and Priya, 2021) – we seek to challenge prevailing stereotypes and assumptions that often marginalize migrant clients' voices within the mental health discourse. This study explores the colonially inflected influences of temporality, professionalism and language on the mental healthcare experiences of migrants in the Netherlands.

Through a series of interviews conducted with migrants and language service providers (LSPs), this research highlights the importance of understanding the lived experiences of these individuals and the systemic barriers they encounter. The findings underscore the need for a paradigm shift in mental healthcare that prioritizes the perspectives and preferences of migrants, advocating for systemic changes that enhance accessibility and equity in care. This study contributes to global mental health literature by highlighting the colonial legacies that undergird systemic barriers faced by migrants, such as cultural misunderstandings and language issues. It presents an underrepresented case in the literature, in the Netherlands, and sheds light on commonly overlooked aspects of the historical context that impacts Europe and the rest of the world. It reevaluates professionalism in mental healthcare by emphasizing the importance of kindness and empathy from the perspectives of migrant patients, positioning them as active participants in their own care. Furthermore, by employing a decolonial framework, this study advocates for actionable systemic changes that promote accessibility and equity in mental health services for marginalized populations.

## Decolonial perspectives and definitions

Coloniality has been defined as "the continuity of colonial forms of domination after the end of colonial administrations, [which] produced colonial cultures and structures in the modern/colonial capitalist/patriarchal world-system" (Grosfoguel, 2007, p. 219). Decoloniality is conceptualized in communication studies as "a critical delinking that offers pluriversal alternatives to modern coloniality" (Enck-Wanzer, 2012, p. 17) and serves as a critical lens through which to examine the structural barriers faced by migrants in accessing mental healthcare (Bhatia and Priya, 2021).

Integrating decolonial perspectives on temporality and effective communication into mental healthcare practices can lead to transformative changes. By challenging the dominant Western narratives that shape mental health interventions, we can develop more culturally responsive and contextually relevant approaches that address the unique needs of migrant populations. For example, in many Indigenous communities, the concept of mental well-being is intrinsically linked to collective healing and connection to land; however, colonial healthcare systems have historically pathologized these beliefs and practices, labeling them as "primitive" or "unscientific" (Gone and Kirmayer, 2020). This colonial legacy continues to impact how mental health services are delivered today, where individual-focused, clinic-based therapy sessions are privileged over communal healing practices or land-based interventions.

Moreover, the concept of temporality is pivotal in understanding the power dynamics between healthcare providers and migrant patients. Temporality, or a perspective on time (Jaszczolt, 2009), varies according to culture (Adam, 2002; Hofstede, 2011; Kastanakis and Voyer, 2014). Postcolonial scholars have argued that time, as it is understood from a Western perspective, imposes a rigid and inflexible value system on oppressed peoples from the Global South (Vazquez, 2009). This tension highlights a power imbalance where providers impose their temporal norms, thereby marginalizing the temporal perspectives of migrants. By recognizing and valuing these diverse understandings of time, we can foster a more inclusive approach to mental healthcare that respects the agency of migrant patients (Hui, 2016).

The concept of professionalism in the medical context is also impacted by colonialism, resulting in present-day inequalities (Evetts, 2013). Not solely characterized by external factors, such as attire or institutional norms, it emerges from the professional's ability to build trust and respect within their community, thus impacting perceptions of professionalism in various contexts (Tadros et al., 2021). Studies suggest that the definition of professionalism can differ across cultures and disciplines, highlighting the need for a consensus on its attributes, particularly in fields like medicine and pharmacy (Monrouxe et al., 2017). Ultimately, a

comprehensive understanding of professionalism, inclusive of diverse sociocultural perspectives, is essential for enhancing practice and ensuring equitable healthcare delivery (AdaAbdel-Razig et al., 2016).

In addition, the historical context of colonialism has significantly shaped contemporary perceptions of migrants, often framing them through a deficit lens that associates low language proficiency with cognitive inferiority (Wekker, 2016; Colak et al., 2023). The linguistic barriers faced by migrants not only hinder their ability to express their mental health needs but also contribute to feelings of disempowerment. Improving access to professional interpreters and acknowledging the preferences of migrants regarding language use can enhance communication and facilitate better healthcare outcomes (Schouten et al., 2022). This aligns with Kirmayer's notion of cultural humility, which emphasizes the importance of understanding the cultural and linguistic contexts of patients (Kirmayer, 2001).

Finally, the call to decolonize mental health is not merely an academic exercise but a necessary step toward reconceptualizing paradigms that perpetuate harm. As the field of global mental health continues to evolve, it is crucial to incorporate the experiences and voices of migrants, ensuring that their needs are met within high-income countries (Millner et al., 2021). By embracing a decolonial framework, we can work toward dismantling the structural barriers that hinder access to mental healthcare for migrants, ultimately promoting health equity and social justice.

## Background and project

The migrant population in the Netherlands includes refugees and individuals from the Global South, many of whom experience heightened mental health challenges due to their backgrounds (Fransen and de Haas, 2022). For example, research in Europe has long shown that those of immigrant background are disadvantaged regarding access to mental healthcare (Lindert et al., 2008). Research in Europe also shows that migrants and those of immigrant background experience higher incidences of psychotic disorders (Anderson et al., 2015), with a higher need for resources that address this specific population (D'Andrea et al., 2023; Termorshuizen and Selten, 2023). In the Netherlands, existing barriers to care include low language proficiency in Dutch, cultural misunderstandings and systemic inequities rooted in colonial histories (Satinsky et al., 2019).

Our multicultural research team conducted a qualitative study as part of the larger European Union-funded "Mental Health 4 All" project, which aims to explore the experiences of migrants in their attempts to access mental healthcare across Europe. To align with the project's goals, we developed a semi-structured interview guide focused on themes including access to care, language barriers and cultural perceptions of mental health. This perspective piece shares valuable insights gained from our interviews with migrants seeking mental healthcare in the Netherlands, contributing to a deeper understanding of their unique challenges and experiences within the broader context of the project.

The interviews were conducted with diverse participants, including 15 third-country nationals (TCNs) and five LSPs representing various sociodemographic backgrounds. It is important to stress that migrants are not a homogeneous group; these participants represent only a portion of the heterogeneous migrant communities in the Netherlands. It is of utmost importance to establish the nuanced differences between migrants, as this paints a more

appropriate picture of the complexities they face (Kiang et al., 2022). Participants were recruited through community organizations and mental health services that cater to migrant populations. The recruitment strategy aimed to include individuals from various cultural backgrounds, ensuring a comprehensive understanding of the challenges faced by these migrants in accessing mental healthcare in this context. For an overview of demographic information, see the Table 1.

Interviews were carried out in participants' preferred languages, including Arabic, Turkish and English, with the assistance of LSPs trained in formal interpretation to facilitate communication. We also interviewed these LSPs at a later time. Each interview lasted ~60 min and was conducted in a private setting to ensure confidentiality and comfort. Participants were informed about the study's purpose and provided informed consent before the interviews began.

**Table 1.** Participants' characteristics

| Pseudonym | Nationality | Gender | Role | Age |
|---|---|---|---|---|
| TCN1 | Lebanese/Syrian | Nonbinary | Mental healthcare seeker | Early 20s |
| TCN2 | Russian | Male | Mental healthcare seeker | Late 30s |
| TCN3 | Egyptian/Lebanese | Transgender female | Mental healthcare seeker | Late 30s |
| TCN4 | China | Female | Mental healthcare seeker | Early 20s |
| TCN5 | Syrian | Male | Mental healthcare seeker | Mid 20s |
| TCN6 | Mauritian | Female | Mental healthcare seeker | Early 30s |
| TCN7 | Hong Kong | Female | Mental healthcare seeker | Early 20s |
| TCN8 | Turkish | Female | Mental healthcare seeker | Mid 40s |
| TCN9 | Turkish | Female | Mental healthcare seeker | Late 40s |
| TCN10 | Moroccan | Male | Mental healthcare seeker | Early 60s |
| TCN11 | Moroccan | Male | Mental healthcare seeker | Mid 30s |
| LSP1 | Turkish | Female | Language service provider | Mid 40s |
| LSP2 | Moroccan | Female | Language service provider | Mid 40s |
| LSP3 | Dutch/French | Female | Language service provider | Mid 50s |
| LSP4 | Dutch | Female | Language service provider | Early 30s |

The interviews were audio-recorded, transcribed verbatim and analyzed using interpretative phenomenological analysis (IPA) (Murray et al., 1999). This approach allowed us to capture the depth of participants' lived experiences and the meanings they ascribed to their interactions with the mental healthcare system (Nizza et al., 2021). The analysis involved initial coding of transcripts, followed by thematic grouping to identify shared patterns and themes among participants' narratives.

To enhance the validity of our findings, we engaged in an iterative process of discussion and reflection within the research team, incorporating diverse perspectives to ensure a comprehensive interpretation of the data. Member checking was also employed, allowing participants to review and provide feedback on the interpretations of their experiences. This collaborative approach aimed to illuminate the complexities of seeking mental healthcare as a migrant in the Netherlands and to inform future practices and policies in this area.

## Thematic analysis: Temporality, professionalism and language

This section presents the findings from the analysis of interviews conducted within the Mhealth4all research project described in the background section, which focused on the experiences of migrants in accessing mental healthcare. Using IPA, we uncovered three themes, that is, temporality, professionalism and language. Each theme sheds light on the complexities and challenges faced by migrants, revealing how historical, cultural and linguistic factors intersect to shape their mental health experiences. The insights gained from this research not only highlight the need for a nuanced understanding of these migrants but also call for a reevaluation of existing mental health practices to better serve diverse populations. Examining these themes, we aim to contribute to the ongoing discourse on decolonizing mental health and fostering more inclusive care systems.

### Temporality: A fluid concept

Temporality, as explored in the interviews, emerged as a fluid concept deeply intertwined with the migrants' lived experiences (Hunfeld, 2022). Many participants described how their past traumas and cultural understandings of time clashed with the rigid, linear timeframes imposed by the healthcare system. For instance, in discussing one of her appointments, one participant stated, "The normal way here is like, everything is slow, slow, slow. It needs to be faster. […] sometime I say, are we meeting like normal time or Dutch time? Dutch time is if it's two o'clock, I need to be 1:55 here and wait 20 min until the person comes. This is the Dutch way." (TCN3). The insights from the interviews suggest that there is a pressing need to reconceptualize temporality in mental healthcare, incorporating flexible scheduling and open-ended intervention models that are sensitive to diverse cultural perceptions of time. This necessitates a departure from one-size-fits-all approaches, fostering an environment where various temporal perspectives can coexist, thereby enhancing the therapeutic alliance and care outcomes for migrant patients.

### Professionalism: Redefining standards

Professionalism has traditionally been framed as an activity in accordance with Western professional standards (Evetts, 2013), often at the expense of alternative healing practices rooted in different cultural backgrounds. Interviews revealed that many migrants felt sidelined by these dominant standards and emphasized the importance of integrating diverse views on professionalism into mental health systems. For example, one participant, in response to a general evaluation from their mental healthcare provider, stated that "I didn't want to hear this. Cause already, this was my friend's words. So, and if this happened for anyone of my friends, I will say this. So I'm a normal person. I'm not a doctor. And I say this, so I don't find it professional." (TCN1). Here, the doctor acted within their professional capacity, representing the institution in a way that is appropriate within the structure of their guidelines (i.e., responding to e-mail, following up with a call, etc.). However, their professional capacities are questioned by the interviewee's redefinition of what constitutes professionalism. In their words, participants "don't find professional" what mental healthcare providers find appropriate, thus delinking a potentially more empathetic interaction from colonially informed measures of professionalism.

Other participants noted that professionalism should be oriented toward kindness, relationality and care. To put it simply, when asked how a mental healthcare provider should be, one participant said, "he should be kind." (TCN5). For another participant, a "nice" doctor meant one who "tries to explain everything until I understand everything" (TCN8). This reevaluation is crucial to ensure that migrants feel respected and understood within the care they receive. Respondents highlighted that practitioners should adopt a balanced approach that honors diverse experiences while still maintaining a foundation of clinical integrity.

### Language: A barrier and a bridge

Language emerged strongly as both a barrier and a bridge in the narratives shared by migrants. The inability to effectively communicate in the dominant language was a significant obstacle, impacting their access to care and the accuracy of their emotional expressions. Participants highlighted the need for mental healthcare providers to adopt a more nuanced understanding of language, moving beyond basic translation to truly comprehend how various languages encapsulate and convey emotional states differently. Recognizing linguistic power dynamics was also vital, as these can affect the therapeutic relationship significantly. The interviews underscored the requirement for culturally and linguistically competent practitioners who can adeptly navigate these differences, ultimately promoting a therapeutic environment that resonates with the diverse experiences of migrant populations. From the practitioner side, one LSP mentioned, regarding Turkish speakers, "Sometimes they are Muslim, sometimes not. So you keep that in the back of your mind." (LSP1). In this interview, the LSP highlights the importance of distinguishing between language, religion, nationality and culture. This recognition is critical in addressing the broader systemic inequalities that persist in access to mental health services, ensuring that all individuals receive the appropriate care they need.

### Recommendations for future research and practice

Navigating the mental healthcare system in the Netherlands presents significant challenges for migrants, particularly regarding access and cultural understanding. This necessitates incorporating migrant perspectives, ensuring that healthcare providers are equipped to understand and respond to their unique experiences (Peñuela-O'Brien

et al., 2023). This knowledge can be used to extend already existing models on the accessibility of healthcare, such as Levesque's model of Patient-centered Access to Health Care (Levesque et al., 2013) or the healthcare access barriers model by Carillo et al. (2011) – for example, through incorporating the nuances of migrants' differing perspectives as valuable input in shaping mental healthcare interventions.

### *Temporality: Raise awareness among mental healthcare providers about differences in cultural understandings of time*

Our research revealed that many migrants faced barriers related to language proficiency and cultural differences in values, temporality in particular, which hindered their ability to effectively communicate their needs to healthcare providers. For instance, the concept of "Dutch time," characterized by strict punctuality and structured scheduling, often clashed with the more fluid perceptions of time held by the migrants in our study. In addition, we found that the lack of culturally competent care often led to frustration and alienation among participants. Therefore, we recommend that organizations working with migrant populations develop training programs for mental healthcare providers that focus on cultural humility and sensitivity, including an understanding of diverse temporal perspectives (Kirmayer and Minas, 2000). By fostering an environment of understanding and respect, healthcare providers can better support migrants in their journey to access mental health services.

### *Professionalism: Implement ongoing support mechanisms for migrants during the healthcare process*

The process of seeking mental healthcare can be unpredictable and overwhelming for migrants, particularly when they encounter systemic barriers (Puthoopparambil et al., 2021). Our findings highlighted the importance of providing ongoing support to migrants throughout their healthcare journey. To address migrant needs in practice, we suggest that mental health organizations establish dedicated support teams trained in cultural power dynamics to assist migrants directly, understanding that they may have differing notions of professional standards due to cultural backgrounds and/or migration experiences. These mediators should possess language skills and an understanding of the cultural contexts of the migrants they serve, including the implications for differing expectations of professionalism. This could involve creating a helpline or in-person support services, staffed where possible by migrants themselves, where migrant patients can receive guidance on accessing care, understanding treatment options and addressing any concerns that arise during their interactions with healthcare providers.

### *Language: Designate trained cultural mediators to facilitate communication and provide clear and accessible information about mental healthcare services*

Effective intercultural communication is crucial in the mental healthcare setting, especially for migrants who may face language barriers. Many participants expressed a need for assistance in navigating the complexities of the mental healthcare system, including understanding their rights and accessing appropriate services. These needs must be taken seriously, as the knowledge migrants have about their own experiences is invaluable in cocreating equitable access to mental healthcare. Our research indicated that many participants struggled to articulate their mental health concerns due to limited proficiency in the local language. This often resulted in misunderstandings and inadequate care. By bridging the communication gap, cultural mediators can help ensure that migrants receive appropriate care tailored to their specific needs, ultimately improving health outcomes (Schinkel et al., 2018).

Access to information about mental healthcare services can vary significantly for migrants, leading to confusion and missed opportunities for care. Our findings revealed that many participants were unaware of the available resources and services, contributing to their reluctance to seek help. To address this issue, we recommend that mental health organizations develop and distribute standardized information sheets that clearly outline the services available to migrants. These materials should be translated into multiple languages – recognizing that not all speakers of a language hail from the same country – and distributed through community organizations, healthcare facilities and online platforms. By ensuring that migrants have access to clear and comprehensive information, organizations can empower them to make informed decisions about their mental health and encourage them to seek the care they need, while also helping them navigate the expectations associated with "Dutch time," professionalism and linguistic differences.

## Conclusions and future perspectives

In summary, the mental health needs of migrants in the Netherlands require urgent attention and a reevaluation of existing care frameworks. By adopting a decolonial perspective, stakeholders can better understand and address the complexities of migrant mental health. A call to action is necessary for all stakeholders to prioritize mental health initiatives, fostering a vision for a future where mental health care is inclusive, equitable and responsive to the needs of all individuals, regardless of their background.

Looking ahead, we believe that future research and practice must prioritize the involvement of marginalized communities in developing mental health services. For example, we advise researchers to involve migrants impacted by coloniality (i.e., refugees and other marginalized groups) throughout the entire research process (e.g., the development of the interview guide and analysis of the data), using participatory research methods such as focus groups and ethnographic observation. This participatory approach can help ensure that care is not only culturally sensitive but also responsive to the unique needs of diverse populations. Future research should also focus on exploring the long-term impacts of culturally competent interventions and the effectiveness of participatory research methods that involve migrants in the research process. Policymakers and healthcare providers are encouraged to prioritize mental healthcare initiatives that address the unique challenges faced by migrants, ensuring that care is accessible, equitable and respectful of cultural differences (Lent et al., 2024).

In addition, we advocate for ongoing critical reflexivity among mental health practitioners, encouraging them to examine their own positionality and how it may influence their practice. In practice, this means practitioners must critically assess how their identities – such as race, gender, socioeconomic status and cultural background – intersect with those of their clients and influence the therapeutic relationship (Dutta, 2010). By engaging in self-reflexivity, professionals can better recognize and mitigate potential power imbalances, avoid imposing their own perspectives and provide more culturally sensitive and equitable care. Positionality, on the other hand, requires an awareness of how one's position in

society and within the therapeutic dynamic shapes their understanding of clients' experiences. Together, these practices foster a more empathetic, inclusive and effective approach to mental healthcare, ensuring that practitioners remain accountable and responsive to the diverse needs of those they serve.

In sum, the mental health needs of migrants in the Netherlands require urgent attention and a reevaluation of the existing care frameworks through a decolonial perspective. Stakeholders must prioritize inclusive and equitable mental health initiatives that respond to the unique needs of all individuals, particularly the marginalized migrant communities. Future research should involve migrants in developing mental health services using participatory methods to ensure cultural sensitivity and responsiveness. In addition, mental health practitioners are encouraged to engage in ongoing critical reflexivity regarding their identities and positionality to provide more empathetic and effective care for diverse populations. The journey toward decolonizing mental healthcare is fraught with challenges, but it is also filled with opportunities for growth and transformation. By embracing a critical perspective, we can work toward a mental health system that truly serves all individuals, honoring their histories and experiences while promoting healing and well-being.

**Open peer review.** To view the open peer review materials for this article, please visit http://doi.org/10.1017/gmh.2025.10038.

**Data availability statement.** The data that support the findings of this study are not publicly available due to ethical restrictions and participant privacy. This research involved qualitative interviews with human participants (migrants and language service providers), and their informed consent explicitly stipulated confidentiality and the protection of their sensitive personal information. In accordance with these ethical obligations and to safeguard participant privacy, the raw interview transcripts, which could potentially contain identifiable information, cannot be made openly accessible. However, anonymized or aggregated data may be made available from the corresponding author upon reasonable request, following institutional ethical review and completion of a formal data sharing agreement. This approach aligns with Cambridge University Press's support for transparency around research materials, while also upholding the critical need to maintain accurate records and respect human participants' rights to privacy.

**Acknowledgments.** The authors would like to acknowledge the participants who willingly participated in the study. Without their willingness, this study would not have happened. In addition, the authors are grateful for the support of the Mental Health 4 All project consortium, including representatives of the following institutions: Universidad de Alcalá, Vrije Universiteit Brussel, Vilniaus Universitetas, Stichting Gezondheid Allochtonen Nederland, Universitätsklinikum Hamburg-Eppendorf, Uniwersytet Warszawski, Univerzita Konštantína Filozofa v Nitre, Università degli studi di Genova, Associatie Marokkaanse Artsen Nederland, European Network for Public Service Interpreting and Translation and Stowarzyszenie na Rzecz Wspierania Psychiatrii Dzieci I Mlodziezy vis-à-vis, Surrey University, University of Amsterdam. Finally, the authors are grateful for the funding of the Asylum, Migration and Integration Fund of the European Union.

**Author contribution.** Gian-Louis Hernandez: Drafted the work and revised it critically for important intellectual content; conducted acquisition, analysis and interpretation of data for the work; provided final approval of the version to be published and agreed to be accountable for all aspects of the work in ensuring that questions related to the accuracy or integrity of any part of the work are appropriately investigated and resolved. Melanie De Looper: Revised the work critically for important intellectual content; conducted acquisition, analysis and interpretation of data for the work; provided final approval of the version to be published and agreed to be accountable for all aspects of the work in ensuring that questions related to the accuracy or integrity of any part of the work are appropriately investigated and resolved. Sabine Braun: Made substantial contributions to the conception or design of the work and agreed to be accountable for all aspects of the work in ensuring that questions related to the accuracy or

integrity of any part of the work are appropriately investigated and resolved. Graham Hieke: Made substantial contributions to the conception or design of the work and agreed to be accountable for all aspects of the work in ensuring that questions related to the accuracy or integrity of any part of the work are appropriately investigated and resolved. Demi Krystallidou: Made substantial contributions to the design of the work and agreed to be accountable for all aspects of the work in ensuring that questions related to the accuracy or integrity of any part of the work are appropriately investigated and resolved. Julia van Weert: Revised the work critically for important intellectual content; provided final approval of the version to be published and agreed to be accountable for all aspects of the work in ensuring that questions related to the accuracy or integrity of any part of the work are appropriately investigated and resolved. Barbara Schouten: Revised the work critically for important intellectual content; provided final approval of the version to be published and agreed to be accountable for all aspects of the work in ensuring that questions related to the accuracy or integrity of any part of the work are appropriately investigated and resolved.

**Financial support.** This work was supported by the Mental Health 4 All project consortium and the Asylum, Migration and Integration Fund of the European Union, Grant Number: 101038491.

**Competing interests.** The authors declare none.

**Ethics statement.** The project (filed as 2022-PC-15420) submitted by Gian-Louis Hernandez complies with the guidelines formulated by the Ethics Review Board of the Faculty of Social and Behavioral Sciences, University of Amsterdam, The Netherlands, and has been approved by the aforementioned Ethics Review Board on September 2, 2022.

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
