## [Reviewer Report]

Thank you for your work on an important topic and for providing a helpful framework to meet the needs of migrants with mental illness. Please see my comments; I hope you find them helpful.

On page 2, The authors talk about systemic inequities rooted in colonial history and cultural misunderstanding without sufficient explanation or providing any examples. They do talk about punctuality in the next section. However, I do not find this convincing; migrants are not a homogenous group, and I do not see punctuality as a Western concept. Besides, the severe mental health challenges these people face could be the main reason for finding it difficult to adhere to the time, or there could be issues related to communication. Doesn’t a flexible schedule framework work best for all sufferers of mental illness? The authors also mention that clients from GS have heightened mental health challenges because of their background without sufficient explanation or evidence.

More information about the participants was needed. As mentioned, migrants are not a homogenous group, and the participants' experiences, backgrounds, and characteristics impact the findings.

It is unclear why the study focused on temporality, professionalism, and language. Were these the findings from the analysis or pre-categories that the researchers focused their analysis on addressing?

The findings' presentation lacks evidence from the interviews. It presents a summary and discussion rather than a proper presentation of the qualitative analysis of the data.

The section on recommendations can apply to all mental health sufferers; there are many generalisations and unpacked terminologies and practices, such as positionality and critical reflexivity. What do they mean in practice? The paper is not balanced as more is given to the recommendations section and little to presenting evidence from the actual interviews. What did the participants suggest to improve the service? Little thought is given to involving migrants in decision-making. There is a mention of community involvement and participatory approaches, but this was not discussed sufficiently in the recommendations.

In general, the paper lacks in-depth engagement with the issue of migrants' mental health and decoloniality. As a reader, I am left wondering who the participants are, how they are different, what their needs, struggles, and challenges are as they describe them, and what their perspectives on the service are and how to improve it.

Thanks for considering my comments. I wish you all the best.

---

## [Reviewer Report]

Thank you for the opportunity to review this interesting manuscript. The current Perspective aimed to provide a decolonial perspective on mental healthcare for migrants in the Netherlands. In order to do so, the authors describe the results of an interview study focused on both migrants and language service providers. Below, I propose some revisions for the manuscript, in particular to restructure and clarify. If these comments are addressed appropriately, I believe the paper can be of relevance to the audience of GMH.

Abstract and impact statement

1. The key concepts that are touched upon in the manuscript (e.g., temporality) are not mentioned in the abstract and statement.

Introduction

2. The introduction could benefit from an explanation of “decolonial framework”: what is this? What does it mean for the design and perspective you aim to provide? Also, the impact statement mentions that the current manuscript provides a decolonial framework. I think it this should also be made more explicit in the introduction itself, not just to contribute to the “ongoing dialogue”.

3. Also, if there is an “ongoing dialogue”, the introduction may benefit from a brief recap of what the current status of the dialogue is/was before the current manuscript. (Have there already been first steps towards a decolonial framework?)

4. The introduction appears to end on a ‘circular’ argumentation. The authors mention that “Through a series of interviews conducted with migrants and language service providers, this research highlights the importance of understanding the lived experiences of these individuals and the systemic barriers they encounter.”, which already describes the results of the study. However, it remains unclear what the rationale was for conducting the interviews. Was it the purpose to provide a new perspective, and was that the reason to conduct the interviews? Or were the interviews part of a larger project and did the analysis possibly lead to new insights that the authors now wish to share in their manuscript?

Background and project (methods)

5. Both the terms “language service providers” (as interviewees) and “interpreters” (to help conducting the interviews) are being used here. Was there a difference between these groups? Or were some of the interpreters perhaps also interviewed as language service providers?

Decolonial perspectives and definitions

6. I would strongly recommend the authors to move the definitions to an earlier part of the manuscript in order to help make the aim of the current perspective more clear. Also, the paragraph on integrating decolonial perspectives etc. could be included earlier in the manuscript, as this would help to make the added value of the current manuscript more clear from the beginning.

7. Can the different concepts described in this paragraph actually be considered ‘decolonial perspectives’ (i.e., are ‘temporality’ and ‘ effective communication’ such perspectives?). The paragraph does not really make these relations explicit, nor mention what exactly is being summed up. Also, were these perspectives a result from the interviews? For temporality, the authors refer to their analyses, but this is not the case for the other themes. The paragraph reads as though it is not a direct result from the interviews (but I am not completely sure).

8. Only a definition for ‘coloniality’, but not for the other concepts is provided in the paragraph, which makes the subheading ‘definition’ a bit redundant. I believe the potential audience may benefit from providing definitions for the other concepts too.

Thematic analysis: temporality, professionalism, and language

9. The first sentence of this paragraph confuses me: “This section presents the findings from a larger research project focused on the experiences of migrants in accessing mental healthcare.” Does that mean also from other studies than the interviews? Then why are these not described under “Background and project”?

10. Ah – I see that the definitions requested under comment #8 are provided in this paragraph. There seems to be quite a bit of overlap in terms of content with the previous paragraph. I would suggest to rethink this structure and perhaps combine the paragraphs together (consecutively describing the themes by first providing the definition, and then a reflection), or first explaining what the general perspectives e.g. from the interviews were and announcing that they will be discussed in further detail in a next paragraph.

11. How did “professionalism” come up in the thematic analysis? Could the authors perhaps give an example in practice?

12. What are the similarities/differences between “language” (in this paragraph) and “effective communication” (previous paragraph)? I would assume that language is perhaps part of effective communication, but that they are not exactly the same. Reflecting on this could possibly strengthen the internal coherence of the manuscript. (As the recommendations again mention “effective communication”).

Recommendations for future research and practice

13. Also related to the internal coherence of the manuscript: how do the specific recommendations relate to the themes from the thematic analysis? It could, for instance, be helpful if there was a clear recommendation for each of the themes.

14. One of the recommendations is to “establish a comprehensive framework” – in what sense is that different from the previously mentioned “decolonial framework”? (If there are no differences, then the impact statement should not mention that the current paper provides such a framework.)

15. Also, the actual recommendation mentioned under this subheading is “fostering an environment of understanding and respect” – in what way does tat relate to the comprehensive framework?

Conclusions and future perspectives

16. Inspired by the impact statement: what, precisely, is the authors’ conclusion regarding the provision of a decolonial framework? Similarly, what exactly is the contribution of the current manuscript to the “ongoing dialogue” that is referred to e.g. also in the abstract? I believe it would be strong to make this explicit as a conclusion.

---

## [Reviewer Report]

Thanks for addressing the previous comments. The paper has improved substantially. However, I still believe it lacks the voice and direct quotes from participants. But if the editor is fine with that, then it is ok. There are still two minor issues to attend to

1- In the background section, according to the letter you provided, which still has comments in the margin, it seems that the following paragraph ( In the Netherlands,

existing barriers to care include low language proficiency in Dutch, cultural

misunderstandings, and systemic inequities rooted in colonial histories (Satinsky et al.,

2019) needs to be amended.

2-In the findings section, when you explain professionalism, the example you provide is not clear. I didn’t understand what the participant was referring to.

Thanks, and wish you all the best.

---

## [Reviewer Report]

Thank you for considering and integrating my prior suggestions to your manuscript. I believe that the structure and clarity are already much improved. However, I have some additional concerns that I would like to see addressed. If these matters are resolved appropriately, I still believe the paper may be of relevance to the audience of GMH.

1. ‘Introduction’: the definition for decoloniality is helpful.

2. ‘Decolonial perspectives and definitions’: because of the improved/restructuring in the background, it has now become much clearer how this project relates to the broader context. However, as per one of my previous comments, it would make more sense to describe the concepts themselves together with the actual finding (i.e., under their specific subheadings in the thematic analysis) by “ consecutively describing the themes by first providing the definition, and then a reflection”, as these concepts were “uncovered” by the interviews. (Although it is later mentioned that specific concepts were “explored in the interviews”. If this is indeed the case – i.e., if you already had these concepts in mind beforehand – the definitions can stay where they are, but I would strongly recommend to specify this in the text.) In case you decide to include the definitions with the specific results, you could e.g. add the description of the decolonial perspectives to the introduction or background.

3. ‘Decolonial perspectives and definitions’: Especially in case you decide to keep this paragraph in its current place: it now reads as a list of different concepts, rather than a comprehensive background/description. For example, the definitions mention ‘the concept of temporality’ and ‘the concept of professionalism’, but also ‘the historical context of colonialism’, but not of them have been properly introduced – they are simply mentioned consecutively. (Also, does the historical context address a similar level concept compared to the other two?) I think this could still be improved by (1) making the relations between the separate paragraphs more clear/explicit, and (2) by introducing the concepts a bit more gradually (e.g. in a ‘funnel’). Also, I think the second paragraph should perhaps be included in the final paragraph of this subheading (explaining the goal of integrating these perspectives, thereby leading to the next paragraph).

4. ‘Decolonial perspectives and definitions’: The example provided in line 18-25 (on the concept of mental well-being) is helpful – can such examples be provided for other concepts too to promote clarity? (Or, as suggested above with comment #2, be integrated as part of the results?)

5. ‘Background and project’: were all of the included “TCNs” (for which a definition is missing) “Mental Healthcare seekers” as indicated in the table? Or were they general migrants/TCNs who may also have had a ‘good’ mental health status? Although recruitment was partly conducted trough mental health services, this is not specified as a criterion. This would be important information to reflect on.

6. ‘Background and project’: Please define the age categories with specific age ranges.

7. ‘Background and project’: The running text mentions that there were five LSPs, but the table only describes four. Please resolve.

8. ‘Thematic analysis’: The results are now much clearer, including their relation to (de)colonialism. However, some English quotes are strangely formulated (e.g. “ I need to be 1:55 here”). Please check and correct.

9. ‘Recommendations for future research and practice’: the link with the result is much more imminent now. However, it now reads as though there may be one major issue/recommendation and that is to account for cultural differences. Specifically, cultural differences are mentioned to be a reason for different notions of time/temporality, for different notions of professionalism, and cultural mediators are mentioned as a solution for intercultural communication. This leads me to question whether temporality/professionalism/language are truly the main concepts/problems at hand that should be resolved, or whether the broader issue are in fact the cultural differences? Can these recommendations not be provided together as one major recommendation (or a general reflection on this)? Or alternatively, can the difference between the different recommendations/resolutions be made more explicit?

10. ‘Conclusions and future perspectives’: the last paragraph reads (unnecessarily) repetitive in addition to the previous three under this heading.

11. General: throughout the manuscript the “ Mental Health 4 All”/“Mhealth4all” project is referred to in different abbreviations. Please resolve this issue.

---

## [Editor Report]

The manuscript has significantly improved following the first review. However, I agree with both reviewers that several key concerns (such as the inclusion of participants' voices, clearer definitions of introduced concepts, and better synthesis of concluding recommendations, among others) still need to be addressed.